# Quantification of Letrozole, Palbociclib, Ribociclib, Abemaciclib, and Metabolites in Volumetric Dried Blood Spots: Development and Validation of an LC-MS/MS Method for Therapeutic Drug Monitoring

**DOI:** 10.3390/ijms251910453

**Published:** 2024-09-27

**Authors:** Eleonora Cecchin, Marco Orleni, Sara Gagno, Marcella Montico, Elena Peruzzi, Rossana Roncato, Lorenzo Gerratana, Serena Corsetti, Fabio Puglisi, Giuseppe Toffoli, Erika Cecchin, Bianca Posocco

**Affiliations:** 1Experimental and Clinical Pharmacology Unit- CRO Aviano, National Cancer Institute, IRCCS, 33081 Aviano, Italy; eleonora.cecchin@cro.it (E.C.); marco.orleni@cro.it (M.O.); sgagno@cro.it (S.G.); elena.peruzzi@cro.it (E.P.); rroncato@cro.it (R.R.); gtoffoli@cro.it (G.T.); bposocco@cro.it (B.P.); 2Doctoral School in Pharmacological Sciences, University of Padua, 35131 Padova, Italy; 3Clinical Trial Office, Scientific Direction- CRO Aviano, National Cancer Institute, IRCCS, 33081 Aviano, Italy; marcella.montico@cro.it; 4Department of Medical Oncology- CRO Aviano, National Cancer Institute, IRCCS, 33081 Aviano, Italy; lorenzo.gerratana@cro.it (L.G.); serena.corsetti@cro.it (S.C.); fabio.puglisi@cro.it (F.P.); 5Department of Medicine, University of Udine, 33100 Udine, Italy

**Keywords:** palbociclib, ribociclib, abemaciclib, letrozole, therapeutic drug monitoring, mass spectrometry, dried blood spot, plasma exposure, C_trough_

## Abstract

Therapeutic drug monitoring (TDM) may be beneficial for cyclin-dependent kinase 4/6 inhibitors (CDK4/6is), such as palbociclib, ribociclib, and abemaciclib, due to established exposure–toxicity relationships and the potential for monitoring treatment adherence. Developing a method for quantifying CDK4/6is, abemaciclib metabolites (M2, M20), and letrozole in dried blood spots (DBS) could be useful to enhance the feasibility of TDM. Thus, an optimized LC-MS/MS method was developed using the HemaXis DB10 device for volumetric (10 µL) DBS collection. Chromatographic separation was achieved using a reversed-phase XBridge BEH C18 column. Detection was performed with a triple quadrupole mass spectrometer, utilizing ESI source switching between negative and positive ionization modes and multiple reaction monitoring acquisition. Analytical validation followed FDA, EMA, and IATDMCT guidelines, demonstrating high selectivity, adequate sensitivity (LLOQ S/N ≥ 30), and linearity (r ≥ 0.997). Accuracy and precision met acceptance criteria (between-run: accuracy 95–106%, CV ≤ 10.6%). Haematocrit independence was confirmed (22–55%),with high recovery rates (81–93%) and minimal matrix effects (ME 0.9–1.1%). The stability of analytes under home-sampling conditions was also verified. Clinical validation supports DBS-based TDM as feasible, with conversion models developed for estimating plasma concentrations (the reference for TDM target values) of letrozole, abemaciclib, and its metabolites. Preliminary data for palbociclib and ribociclib are also presented.

## 1. Introduction

The use of therapeutic drug monitoring (TDM) has been proposed for cyclin-dependent kinase 4/6 inhibitors (CDK4/6is) such as palbociclib, ribociclib, and abemaciclib [1,2,3]. These CDK4/6is are oral targeted drugs approved for the treatment of hormone receptor (HR)-positive, human epidermal growth factor receptor 2 (HER2)-negative locally advanced or metastatic breast cancer in combination with an aromatase inhibitor (letrozole/anastrozole/exemestane) or the selective estrogen receptor degrader (SERD), fulvestrant [4,5,6,7,8,9].

CDK4/6is meet several criteria for TDM suitability [10]. Palbociclib, ribociclib, and abemaciclib have narrow therapeutic indexes and show significant inter-individual variability in plasma exposure, with coefficients of variation for minimum steady-state plasma concentration (C_trough_) ranging from 40 to 95% [11]. Exposure-response (E-R) analyses for toxicity have demonstrated a positive relationship between drug exposure and toxicity, particularly neutropenia [7,8,9]. These findings were recently confirmed by a study linking ribociclib C_trough_ with neutropenia and QT prolongation [12]. Moreover, abemaciclib showed a positive exposure-efficacy relationship in the MONARCH studies, with drug plasma levels affecting both PFS and tumor size reduction [7,13], while no clear E-R trend for efficacy has been reported for palbociclib and ribociclib [8,9,12,14]. To note, the recent prospective multicenter DPOG-TDM study provided a proof of concept of the feasibility and potential clinical utility of an individualized dosing based on TDM for palbociclib [15]: 62% of patients (13 out of 22) were underdosed (using the mean population C_trough_ of 61 ng/mL as the TDM target), and the pharmacokinetic-guided intervention allowed to obtain an adequate exposure in four of five assessable patients (80%). An additional benefit of a clinical application of TDM of CDKis in clinics is the possibility to monitor the patients adherence to therapy, a prerequisite for efficacy [16]. Encouraging persistence through patient education and drug monitoring can improve adherence [10,17,18].

From an analytical perspective, the use of dried blood spot (DBS) samples (also known as microsampling) significantly enhances TDM feasibility in clinical practice [19,20]. Compared to traditional venous blood sampling, DBS methods are minimally invasive, can be performed by patients at home, and allow simplified pre-analytical steps (DBS samples are often more stable at room and high temperatures, making their transport and storage less expensive, and present a lower biohazard risk due to the dried material) [19]. Nonetheless, quantitative results of DBS can be affected by the hematocrit (Hct), influencing pre-analytical, analytical, and post-analytical phases [19,20]. The Hct value influences the pre-analytical step as it determines differences in blood spreading on the paper, resulting in drops of different sizes at different Hct values (blood with a high Hct will spread less than blood with a lower Hct due to the differences in viscosity of the blood). With conventional DBS (collection of a nonvolumetric drop of blood, free falling, or by touching onto a filter paper directly from a finger prick), a fixed diameter sub-punch is taken from the spot, and the difference in spreading leads to a difference in sample proportion. Although it has a minor impact, it also affects the analytical phase by influencing the recovery of the analyte (recovery is usually lower when the Hct value is higher) and the matrix effect (sample with a different Hct can be considered to be a different matrix). The use of volumetric DBS (analysis of the whole punch after volumetric application of a fixed blood volume) substantially reduces the Hct effect with the removal of the Hct-based area bias (differences in blood spreading due to varying Hct) [21]. This issue is particularly pertinent in cancer patients, whose Hct values can vary widely [22,23,24]. The reduction of Hct impact on volumetric DBS quantification also simplifies the conversion between venous plasma and capillary DBS (post-analytical phase), which is necessary since reference intervals for most anticancer drugs are plasma-based [25].

Currently, one LC-MS/MS method for quantifying palbociclib, ribociclib, and abemaciclib in volumetric DBS samples using volumetric absorptive microsampling (VAMS) is published [26]. As this method was developed for treatment adherence monitoring, no attempt was made to evaluate blood-to-plasma conversion. However, observed blood-to-plasma concentration ratios were not reproducible, indicating quantification in finger prick blood sampled by VAMS and plasma cannot be used interchangeably.

The aim of this study was to develop an LC-MS/MS method for quantifying these drugs using the HemaXis^®^ device (DBS System SA, Gland, Switzerland) as an alternative to VAMS for sample collection and quantify letrozole and abemaciclib main metabolites in addition to the three CDK4/6 inhibitors. In fact, abemaciclib is extensively metabolized by CYP3A4 with the formation of two active and significantly abundant metabolites, N-desethyl and hydroxyl abemaciclib (M2 and M20, respectively). M2 and M20 should be considered for monitoring as the areas under the plasma concentration-time curve (AUCs) represent 39% and 77% of that of the parent compound, respectively [4,11]. In addition, our previous study showed high variability in the relative abundance of M2 and M20, emphasizing the need to monitor the active metabolites together with abemaciclib [24,27]. TDM of letrozole is also suggested, as exposure (C_trough_ ≥ 85.6 ng/mL) was associated with a longer time to progression (TTP) [28]. Efforts were made in the optimization of sample treatment to reduce the Hct-based recovery bias to finally obtain an Hct-independent and reliable quantification method.

## 2. Results

Quantification of letrozole, palbociclib, ribociclib, abemaciclib, and their main metabolites (M2 and M20) in DBS samples using the HemaXis DB10 device was conducted using chromatographic and mass spectrometric parameters optimized from our previously published LC-MS/MS method [27]. This method, originally developed for the quantification of the same analytes in plasma samples, was employed to establish reference plasma concentrations for the clinical validation study of the DBS-based method. Building on the previously optimized LC-MS parameters, a specific DBS extraction procedure was established to achieve Hct-independent quantification, followed by comprehensive analytical and clinical validation studies.

### 2.1. Preparation of Calibration Curve, Quality Controls, and Patient Samples

A preliminary estimation of the Hct effect on quantification was performed during the evaluation of the optimal sample preparation method. As detailed in Section 3.3, Hct independence was achieved through a two-step preparation process. This involved an initial addition of 200 µL of ultrapure water, mixed for 30 min, followed by protein precipitation with 600 µL of ISs solution (acetonitrile). As anticipated due to the dilution effect and as illustrated in Appendix A, this procedure resulted in lower signal intensity compared to other methods. To enhance signal intensity, the final dilution step with aqueous mobile phase (MPA, pyrrolidine-pyrrolidinium formate buffer, 5:5 mM, pH 11.3) was adjusted, reducing the dilution factor from 2 (used during the experiment) to 1.5 without compromising the chromatographic performance. Additionally, the injection volume was set to 20 µL. As extensively detailed in Section 2.3.1, these modifications ensured that the signal-to-noise (S/N) ratio of the lower limit of quantification (LLOQ) samples was more than acceptable for quantification purposes.

### 2.2. LC-MS/MS Method

Adequate separation of analytes (resolution Rs > 2) and peak symmetry were achieved. Figure 1 shows representative SRM chromatograms of blank DBS (a), blank DBS with addition of ISs (b), LLOQ sample (c), and DBS from patients treated with abemaciclib, palbociclib, ribociclib, and letrozole (d–f). The following retention times were obtained for the analytes: 4.08 min for letrozole, 5.38 min for ribociclib, 6.10 min for palbociclib, 6.80 min for M2, 7.02 min for M20, and 7.46 min for abemaciclib (Figure 1c–f).

### 2.3. Analytical Validation

#### 2.3.1. Selectivity, Sensitivity, and Linearity

The method proved to be selective, as no interfering peak was observed at the retention times of each analyte and internal standard (IS) (Figure 1a). The LLOQ signal showed an S/N ranging from 30 (for M20) to 250 (for abemaciclib) (Figure 1c), with a precision of ≤14.9% and an accuracy between 87 and 112% for all the analytes (Appendix A); these values were within the acceptance sensitivity criteria.

Good linearity was demonstrated by the Pearson correlation coefficient (r) obtained for each analyte, which were ≥0.997. This result was also confirmed by the accuracy and precision results of the calibration curves: accuracy ranged from 97 to 103% for all compounds, while precision was ≤9.1% (Appendix A).

#### 2.3.2. Carryover

As observed in our previously published method [27], analysis of the first blank sample injected after the upper limit of quantification (ULOQ) sample revealed no carryover effect, as no quantifiable peaks for any analytes (including ISs) were detected.

#### 2.3.3. Accuracy and Precision

The within-run accuracy and precision results for each working day are reported in Appendix A. Taking into account all the analytes, accuracy ranged from 87 to 112%, with a CV% of ≤14.9%. The between-run accuracy and precision data are presented in Table 1, showing accuracy ranging from 95% to 106% with a CV% of ≤10.6%. These results demonstrate satisfactory accuracy and precision for the tested concentration levels and analytes.

#### 2.3.4. Haematocrit Effect

According to Table 2, the proposed method demonstrates Hct independence within an Hct range of 22–55%. The accuracy and precision of the LLOQ and quality control (QC) samples consistently met the acceptance criteria. Samples with an Hct of 22% showed an accuracy range of 94–111% and a CV% of ≤12.5%. Samples with an Hct of 55% exhibited an accuracy range of 93–104% and a CV% of ≤12.1%. In comparison, our previous method for quantifying letrozole, palbociclib, and ribociclib in non-volumetric DBS [24] was Hct-independent within a slightly narrower range of 25–49% and demonstrated an accuracy range of 88–115%.

#### 2.3.5. Recovery and Matrix Effect

According to Appendix A, the recovery was comparable among the different analytes and between low and high concentrations, ranging from 81 to 93%, and it was consistent among the different matrix sources, with the CV% always ≤12.3%. The optimized sample treatment for extracting analytes from the DBS resulted in higher recoveries compared to our previous non-volumetric DBS-based method [24], which used Whatman ET31CHR as filter paper and showed a recovery range of 68 to 80%. No significant matrix effect was detected among the six different donors and the low (22%) and high (55%) Hct values. The calculated ME was in the range of 0.9–1.1% with a CV% of ≤8.9% for all the analytes, while the measured QC normal showed an accuracy ranging from 96 to 99% and a CV% of ≤6.8% (Appendix A).

#### 2.3.6. Stability

As shown in Appendix A, analytes were stable after extraction and dilution in an autosampler (5 °C) for 5 days, and the DBS samples can be stored in the desiccator (20 °C, humidity <35%) for 11 months without compromising the measurements. It also seemed that home sampling is feasible since analytes were stable after two weeks in a plastic bag containing two silica gel packets followed by 7 weeks of storage in a Sicco desiccator, with the final accuracy between 88 and 106% and a CV% of ≤ 4.3% (Appendix A).

### 2.4. Clinical Validation

For clinical validation, a minimum number of 40 samples from at least 25 subjects is recommended, as outlined in the guideline by Capiau et al. [29]. Currently, only a limited number of paired plasma and DBS samples (28) have been collected from patients. A preliminary statistical evaluation was performed on 20 samples for abemaciclib (and its metabolites) and 22 samples for letrozole. The patients, all female with a mean age of 55 years (range 36–81), were affected by breast cancer and were treated with abemaciclib, palbociclib, or ribociclib, mostly in combination with letrozole. Detailed therapeutic schemes of the included patients are provided in Appendix A.

As described in Section 3.6, plasma concentration was estimated (ECpla) using the following equation: ECpla = CDBS/CF, where the CF (correction factor) was calculated as the average ratio between the CDBS (DBS concentration) and the Cpla (plasma concentration): CF = CDBS/Cpla. The CF values determined for each analyte are listed in Table 3. The data for letrozole, palbociclib, and ribociclib largely correspond to those obtained using our previous method for quantifying these analytes in non-volumetric DBS [24]. Since the CF corresponds to the plasmatic fraction (Fp), a CF close to 1 indicates that the drug is evenly distributed between the red blood cells and the plasma fraction. This is the case for abemaciclib, where CDBS can be considered equivalent to Cpla without further conversion. For letrozole and M20, where the CF is slightly lower than 1, a conversion is necessary. The literature supports our findings, with 35.2 ± 2.7% of letrozole found in erythrocytes at an Hct of 0.4 [30]. In contrast, the distribution behavior of palbociclib, ribociclib, and M2 is different, as the CF (i.e., Fp) value is higher than 1, indicating these drugs are primarily distributed in the erythrocytes rather than in the plasma.

The application of the calculated CF to the DBS measurements allowed the estimation of plasma concentrations in most samples. Even with the poorest predictive performance observed for the abemaciclib metabolite M2 (% equivalence of 75%), the equivalence criteria required by the guidelines were met. For the other analytes, the percentage of equivalence ranged from 80% to 100% of the samples (Table 3). Figure 2 displays the percent difference of ECpla relative to actual Cpla for each analyte. Statistical analyses were performed only on letrozole and abemaciclib, with results listed in Table 4. Satisfactory agreement between ECpla and Cpla was demonstrated for each analyte (Figure 3). Bland–Altman plots showed a very low absolute mean bias (≤6.2 ng/mL) and no statistically significant correlation between the difference (ECpla–Cpla) and the mean of the values (ECpla and Cpla), with the highest ρ being −0.322 (*p* ≥ 0.166). The slope values of the Passing–Bablok regression were close to 1 (ranging from 0.8 for M2 to 1.0 for letrozole), with the 95% CI always including 1. The 95% CI of the intercept included 0 in all cases. Lin’s concordance correlation coefficient confirmed the good agreement, with values between 0.8850 for M2 and 0.9901 for letrozole. Compared to the non-volumetric DBS method [24], the use of the HemaXis DB10 device for volumetric sampling of the blood drop increased the agreement between ECpla and Cpla of letrozole, raising the percent of equivalence from 87% to 100% (with percent difference within ±10%). No further comments can be made regarding palbociclib and ribociclib, although a DBS-to-plasma conversion method based on a simple CF seems feasible. This represents an improvement over the non-volumetric DBS method, where the best percent equivalence between ECpla and Cpla of palbociclib was achieved using a formula that included both the drug’s partition coefficient and the patient’s Hct.

### 2.5. Incurred Sample Reanalysis

A total of 14 patient samples were analyzed twice for the assessment of method reproducibility. Excluding two samples for M2, all the calculated percent differences were within ±20% (ranging from 18 to −14%), as shown in Figure 4.

## 3. Materials and Methods

Quantification of analytes in DBS using the HemaXis^®^ DB10 device (DBS System SA, Gland, Switzerland) was performed applying chromatographic and mass spectrometric parameters optimized for our previously published LC-MS/MS plasma-based method [27]. Here, we describe the DBS-specific validation study and all the details that differ from the previous method.

### 3.1. Chemicals and Reagents

Characteristics of reference materials and solvents used are comprehensively reported in the published paper [27]. In brief, abemaciclib (purity ≥ 99.7%) and the labeled internal standard (IS), D8-abemaciclib (2H purity 98.7%), were purchased from Clearsynth (Maharashtra, India). Palbociclib (purity 100%), ribociclib hydrochloride (purity ≥ 99.5%), and the ISs, D6-ribociclib (2H purity 99%), D8-palbociclib (2H purity 98.3%), and 13C2,15N2-letrozole (13C purity 99.6%, 15N purity 99.6%) were synthesized by Alsachim (Illkirch Graffenstaden, France). The M2 (CAS N 1231930-57-6, purity 99.4%) and M20 (CAS N 2138499-06-4, purity 98.2%) metabolites and letrozole (purity 100%) standards were supplied by MedChemExpress (Monmouth Junction, New Jersey, USA). Acetonitrile (hypergrade for LC-MS) was obtained from Merck (Darmstadt, Germany). Drug-free K_2_EDTA human whole blood from healthy volunteers was collected at the Transfusion Unit of our institution and used to prepare the calibration curves and quality control samples (QCs). Whatman 903 paper was purchased from GE Healthcare (Westborough, MA, USA); finger prick blood samples were collected using Accu-Check Safe-T-Pro Plus lancets (1.8 mm penetration depth, 0.63 mm gauge) from Roche Diagnostics (Mannheim, Germany).

### 3.2. Preparation of the Standard Solutions

Stock and working solutions (WSs) of each analyte were prepared as previously reported [27]. Briefly, stock solutions of the analytes (Appendix A), prepared in duplicate, were mixed and diluted with MeOH to obtain the WSs for both the calibrators (eight points, from A to H) and QC (H—high, M—medium, L—low), with the concentrations listed in Appendix A. The stock solutions of each internal standard (IS) (prepared at 1 mg/mL each) were mixed and diluted with acetonitrile to obtain a solution with the following composition: 15.5 ng/mL for D8-palbociclib and ^13^C_2_,^15^N_2_-letrozole, 54 ng/mL for D6-ribociclib, and 24 ng/mL for D8-abemaciclib.

### 3.3. Preparation of Calibration Curve, QCs, and Patient Samples

DBS calibrators and QC samples were prepared as follows: 10 µL of the corresponding WS was spiked into 190 µL of human whole blood with the Hct adjusted to 36% using the plasma removal/addition procedure [31]. This Hct value corresponds to the mean found in our patient setting [24]. The samples were gently mixed and incubated at 37 °C for 30 min at 300 rpm in a Vortemp 56 (Illumina, San Diego, CA, USA). After incubation, 10 µL of the mixture was spotted onto Whatman 903 paper, dried, and stored at room temperature in a Sicco Star desiccator (Bohlender, Grünsfeld, Germany) with a humidity level below 35% until analysis. The final concentrations of the calibrators and QC samples are listed in Appendix A.

Finger-prick DBS samples were collected from patients following the instruction of the HemaXis^®^ DB10 device [32] using Accu-Check Safe-T-Pro Plus lancets (1.8 mm penetration depth, 0.63 mm gauge) from Roche Diagnostics (Mannheim, Germany). The HemaXis^®^ DB10 device uses a microfluidic chip with a standard filter card (Whatman 903) to obtain DBS samples with a fixed volume of 10.0 ± 0.5 μL [19,32]. Drying and storage conditions were the same applied to calibrators and QCs. Calibrators, QCs, and patients’ DBS samples were punched with an 8-mm-diameter manual puncher, allowing for volumetric sampling (10 µL). The punched paper disks were transferred to a 2-mL polypropylene (PP) tube with the addition of 200 µL of ultrapure water and mixed using a microplate shaker (VWR, Radnor, TN, USA) for 30 min at 20 °C and 500 rpm. Subsequently, 600 µL of ISs solution were added, and samples were vortexed and centrifuged (17,000 g for 10 min at 4 °C). Finally, 100 µL of the supernatant was diluted with 50 µL of aqueous mobile phase (MPA), transferred to a PP vial, and stored at 5 °C in an LC autosampler until analysis. Figure 5 shows the DBS sample process from collection to analyte extraction. 

Several other procedures for analyte extraction from the punched paper disks were tested: (1) addition of 250 µL of methanol; (2) addition of 250 µL of methanol with 0.1% of NH_4_OH; (3) addition of 250 µL of H_2_O:methanol mixture (20:80, *v*:*v*); (4) addition of 250 µL of H_2_O:acetonitrile mixture (30:70, *v*:*v*). Each condition was followed by 30 min of mixing and final dilution with MPA. This experiment was performed with DBS prepared at QCM concentration and in quintuplicate at each Hct value (22%, 36%, 55%). The extraction procedure was considered adequate if the mean peak area was reproducible between the different Hct levels. Procedures 1, 2, and 4 were excluded because they did not allow Hct independence of the quantification over the tested range of 22–55% (Appendix A). Treatment 3 was excluded because the extract solution was quite dirty (the obtained solution was light red).

### 3.4. Chromatographic and Mass Spectrometric Conditions

As previously reported [27], the chromatographic separation was achieved with a SIL-20AC XR autosampler coupled with LC-20AD UFLC Prominence XR pumps (Shimadzu, Tokyo, Japan) using an XBridge BEH C18 (2.5 μm 3.0 × 75 mm XP) column and a Security Guard Cartridge (XBridge BEH C18, 2.5 µm, 2.1 × 5 mm) from Waters (Milford, MA, USA). Separation was performed with a multi-step gradient composed by a pyrrolidine-pyrrolidinium formate buffer (5:5 mM, pH 11.3) (MPA) and MeOH (MPB) with a total run time of 15.5 min. The use of this buffer as MPA was introduced to improve the peak symmetry, and indeed, the tailing factor decreased from 2 with a pH of 9 (as previously observed [33]) to less than 1.4 for all analytes. The needle wash solution consisted of acetonitrile-water (70:30 *v*/*v*). The chromatographic gradient used was as follows: 30% MPB (methanol) for 0.5 min (initial conditions), increased to 90% MPB in 7 min, 90% MPB maintained for 1.5 min (washing), decreased to 30% MPB for 0.5 min, and 30% MPB maintained for the re-equilibration step (6 min). The oven temperature was set at 45 °C, and the injection volume was 20 µL.

Analytes were ionized and detected using a triple quadrupole API 4000 (AB SCIEX, Framingham, MA, USA) equipped with a TurboIonSpray source operating in negative ion mode for detection of letrozole and its IS and in positive mode for all the other analytes and corresponding ISs. The negative mode was necessary for the detection of letrozole and its IS because the use of a basic (pH 11.3) MPA drastically reduced the signal intensity of these analytes. The optimized source- and compound-dependent parameters are summarized in Appendix A. Data processing and quantification of the analytes were performed using Analyst software (version 1.6.3). Acquisition was performed in scheduled MRM mode using the following transitions (Figure 6): *m*/*z* 284 > 242 for LETRO; *m*/*z* 288 > 246 for 13C2,15N2-letrozole; *m*/*z* 507 > 393 for abemaciclib; *m*/*z* 523 > 409 for M20; *m*/*z* 479 > 393 for M2; *m*/*z* 515 > 393 for D8-abemaciclib; *m*/*z* 448 > 380 for palbociclib; *m*/*z* 456 > 388 for D8-palbociclib; *m*/*z* 435 > 322 for ribociclib; and *m*/*z* 441 > 373 for D6-ribociclib. Metabolites M2 and M20 were quantified using D8-abemaciclib as an IS. The chemical structures of the ISs are shown in Appendix A.

### 3.5. Analytical Validation

The analytical validation was conducted in accordance with the FDA and EMA latest guidelines [34,35] and the Official International Association for Therapeutic Drug Monitoring and Clinical Toxicology Guideline [29] for DBS-specific assessments.

#### 3.5.1. Selectivity, Sensitivity, and Linearity

To evaluate the presence of potential interfering substances, 6 blank samples, each obtained from a different healthy volunteer, were analyzed. The method was considered selective if no peak attributable to interfering components is observed at the retention times of the analytes or the ISs in the blank samples with a peak intensity of ≥20% of the analyte response and ≥5% of the IS response at LLOQ level.

The method sensitivity was evaluated by analyzing LLOQ samples prepared in quintuplicate in three runs (during the within and between accuracy and precision evaluation; see Section 3.5.3). The LLOQ response should be 5 times higher than that of the zero samples and have a signal-to-noise (S/N) ratio of ≥10; precision should be ≤20%, while accuracy should be between 80 and 120% for at least 67% of the samples.

Calibration curves were generated using eight non-zero calibration standards (concentrations are listed in Appendix A). Linearity was evaluated using two different sets of eight calibrators in 5 analytical runs corresponding to 5 working days. A linear regression model with a weighting factor of 1/x^2^ was applied. Accuracy, obtained as back-calculated concentrations, of each calibrator should be between 85 and 115% (80–120% for the lower limit of quantification, LLOQ). The precision, calculated as CV%, should be ≤15%.

#### 3.5.2. Carryover

With the exception of letrozole, the presence of carryover effects was found for all analytes in previously published methods in plasma matrix [36,37,38,39]. This phenomenon was assessed by injecting extracted blank DBS samples after analyzing the upper limit of quantification (ULOQ). The residual signals of the analytes in the blank DBS should be <20% of the LLOQ and <5% of the IS.

#### 3.5.3. Accuracy and Precision

Accuracy and precision were evaluated using LLOQ, QCL, QCM, and QCH samples prepared in quintuplicate. These samples were analyzed within a single working day (within-run) and across three different analytical runs on separate days (between-run). The measured concentrations needed to be within ±15% of the nominal value (accuracy range: 85–115%) with a CV% of ≤15% for 67% of the samples at each concentration level. For LLOQ samples, the accuracy range was set to 80–120%, with precision ≤ 20%.

#### 3.5.4. Hematocrit Effect

Although the whole punch analysis after volumetric application of a fixed volume of blood can nullify the Hct-based area bias, the effect of Hct on recovery and matrix effect could still be an issue. Both the Hct-based recovery bias and the Hct-based matrix effect bias can affect the precision and accuracy of the analytical result. For this reason, the effect of Hct on analyte quantification was evaluated using LLOQ, QCL, QCM, and QCH samples prepared in sextuplicate at Hct levels of 22% and 55%. These samples were quantified using a calibration curve prepared with blood at an Hct of 36%. The impact of Hct was assessed by evaluating the accuracy and precision of the measurements.

#### 3.5.5. Recovery and Matrix Effect

Recovery and matrix effect evaluations were conducted on samples obtained from six different female donors plus two blood samples with artificial Hct levels of 22% and 55% for a total of eight different matrix sources. The analysis was performed in triplicate for low and high QC. Recovery was evaluated by comparing the absolute response (peak area) of the blank DBS extract to which the analyte had been added after extraction procedure (QC post-extraction) with the absolute response of an extract of DBS to which the same amount had been added before extraction procedure (QC normal): QC normal/QC post-extraction × 100. The matrix effect (ME) was determined as the ratio between the absolute response of the QC post-extraction and the same amount of analytes in neat solvent (QC in solvent):QC post-extraction/QC in solvent. The CV% of the ME should not exceed 15%. In addition, ME was assessed through the evaluation of the accuracy and precision of the QC normal quantified using a calibration curve at Hct of 36%.

#### 3.5.6. Stability

The stability of analytes was assessed under various conditions using three replicates of both QCL and QCH. Autosampler stability of the final extract was evaluated until after five days at 5 °C. Long-term stability was tested in a desiccator (Sicco Star) at 20 °C with humidity maintained below 35%. To simulate the conditions of home sample collection by patients, freshly prepared QCL and QCH DBS samples were dried at room temperature for one night and stored (at room temperature) in plastic bags with two 1 g silica gel packets for two weeks, then transferred to the Sicco for an additional 3 and 7 weeks before analysis. Stability was confirmed if accuracy and precision met the acceptance criteria.

### 3.6. Clinical Validation

Since TDM targets are generally reported in the literature as plasma concentrations, the implementation of DBS samples requires the stimation of plasma concentration based on the DBS measurement using a conversion strategy [25]. To develop a conversion model, paired plasma and DBS samples were necessary. Samples were collected from patients participating in a clinical trial (protocol ID: CRO 2022-14, approval date: 12 April 2022) conducted at our institution and approved by the local ethics committee (Comitato Etico Unico Regionale—CEUR). The study was conducted in accordance with the principles of the Declaration of Helsinki, and written informed consent was obtained from all participants. For all participating patients, data on Hct value, dates of Hct measurement, time of last intake, and treatment compliance were recorded. After reaching steady state, paired venous blood and DBS samples were collected within ±30 min of each other using the HemaXis DB10 device. Plasma concentrations, which served as reference points for evaluating the feasibility of the DBS-based method, were obtained using the previously validated LC-MS/MS method [27] (see Section 3.7).

We explored the feasibility to obtain the estimated plasma concentration (ECpla) based on a correction factor (CF), calculated as the average ratio between the concentration in DBS (CDBS) and the concentration in plasma (Cpla): CF = CDBS/Cpla. This CF for each analyte was then applied to obtain ECpla from the DBS measurement as follows: ECpla = CDBS/CF. Statistical analysis such as Passing–Bablok regression analysis, Bland–Altman plots, and Lin’s concordance correlation coefficient (Lin’s CCC) were performed with STATA 14.2 software (StataCorp, Lakeway Drive, College Station, TX, USA) to evaluate the agreement between ECpla and actual Cpla. For Passing–Bablok analysis, the intercept and slope of the regression equation are reported with a relative 95% confidence interval (95% CI); for Bland–Altman plots, bias is reported with a 95% CI. The Spearman correlation coefficient (ρ) between (Y − X) and (X + Y)/2 is calculated (where Y represents ECpla and X represents Cpla), and a p-value < 0.05 was considered statistically significant. Moreover, according to EMA and FDA guidelines [34,35], the difference between ECpla and Cpla (% difference) should be within ±20% in at least 67% of the samples analyzed, calculated as follows: %difference = (ECpla − Cpla) × 100/Cpla. In this equation, the difference (ECpla − Cpla) is divided by Cpla and not by the mean of the two measurements, as this is the reference. The percentage of samples that showed a % difference within ±20% was defined as percentage equivalence (% equivalence).

### 3.7. Plasma Sample Quantification

Plasma concentrations were obtained using a bioanalytical method previously developed and validated in our laboratory [27]. The assay was characterized by the same analytical range and LC-MS setup. Analyte extraction from the plasma matrix was conducted as follows: calibrators and QCs were prepared by spiking WS into the plasma at a ratio of 1:20, while patients plasma samples were thawed at room temperature. Then, 50 µL of spiked plasma/patient sample was transferred to a 1.5-mL tube, and 150 µL of methanol containing the ISs was added for protein precipitation. The samples were vortex-mixed for 10 s, then centrifuged (16,200× *g* for 10 min at 4 °C). Finally, 60 µL of the supernatant was diluted with 140 µL of MPA (pyrrolidine-pyrrolidinium formate buffer 5:5 mM, pH 11.3), vortexed, and centrifuged (16,200× *g* 10 min at 4 °C). One hundred and fifty-µL of the resulting solution was transferred to a polypropylene vial and stored at 15 °C in an LC autosampler until analysis.

### 3.8. Incurred Samples Reanalysis

A subset of samples for each CDK4/6i was analyzed twice in two runs to provide an additional assessment of the robustness and reproducibility of the method according to the EMA and FDA guidelines (incurred sample reanalysis, ISR). The two analyses were considered equivalent if the percent difference (ISR% difference) between the two measured concentrations, calculated using the formula: ISR% difference = [(repeat − original) × 100/mean(original − repeat)], was within ±20%.

## 4. Conclusions

In this study, a robust LC-MS/MS method for quantifying letrozole, palbociclib, ribociclib, and abemaciclib and its metabolites in volumetric dried blood spots (DBSs) was successfully developed and validated. The proposed method, utilizing the HemaXis DB10 device, was found to be independent of Hct variations and showed excellent selectivity, sensitivity, linearity, accuracy, and precision across the tested ranges. The specific DBS extraction procedure effectively minimized the Hct effect, which was shown to be negligible within the range of 25–55%, with accuracy between 94 and 111% at 22% Hct and 93–104% at 55% Hct. Additionally, this method achieved high analyte recovery from the punched disc, with percent recovery ranging from 81% to 93%. The results confirm that this method is suitable for TDM of these compounds in a clinical setting, facilitating personalized dosing strategies. The stability of analytes under home-sampling conditions was assessed: letrozole, abemaciclib and its metabolites, palbociclib, and ribociclib were stable for two weeks when stored at room temperature in plastic bags containing two 1 g silica gel packets. They can subsequently be stored in a Sicco desiccator (20 °C, humidity <35%) for seven weeks until analysis. Based on our previous experience, calibration curves were constructed to encompass a wide range of clinically meaningful concentrations for each analyte. This approach allowed the quantification of 28 DBS samples collected from participating patients. It is well known that the correlation between plasma and DBS drug concentrations requires specific normalization, which is analyte-dependent. Despite the limited number of samples tested, it was evident that a DBS-to-plasma conversion was needed for all analytes except for abemaciclib, which showed a CF (corresponding to the plasmatic fraction) equal to 1. In this case, quantification of abemaciclib in plasma or DBS samples yields the same value. Although preliminary, the clinical validation of letrozole and abemaciclib indicated the feasibility of estimating plasma concentration from DBS measurements. The method’s compatibility with DBS samples, coupled with its minimized Hct effect, offers a practical and reliable approach for clinical implementation, potentially improving patient adherence and treatment outcomes in hormone receptor-positive, HER2-negative breast cancer therapy.

## Figures and Tables

**Figure 1 ijms-25-10453-f001:**
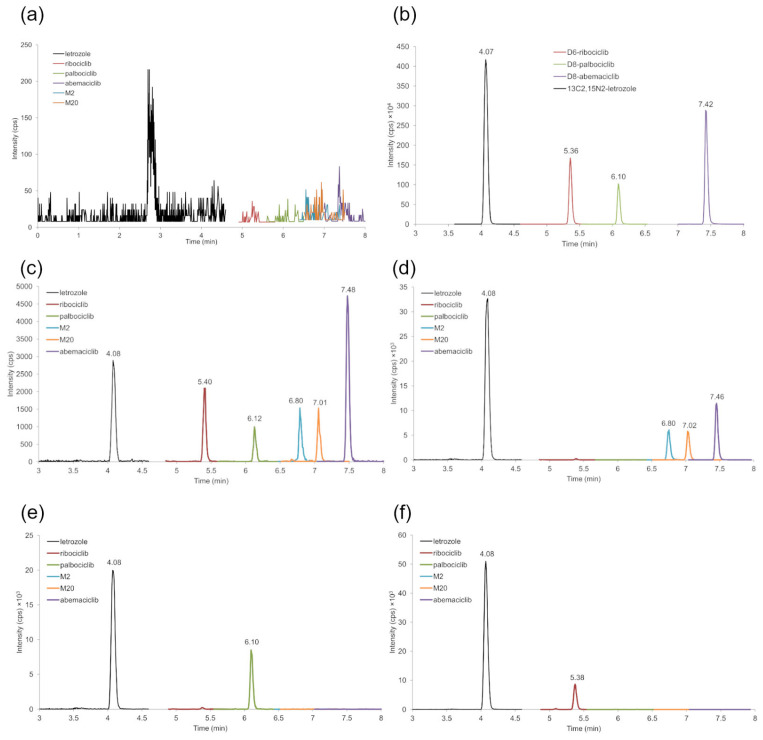
Representative SRM chromatograms of a blank DBS sample (**a**), a blank DBS sample containing the ISs (**b**), an LLOQ sample (**c**), and DBS samples from patients treated with abemaciclib (**d**), palbociclib (**e**), and ribociclib (**f**). The measured concentrations were 110 ng/mL for abemaciclib, 86 ng/mL for M20, 104 ng/mL for M2, and 84 ng/mL for letrozole, 58 ng/mL for palbociclib and 50 ng/mL for letrozole, 653 ng/mL for ribociclib, and 114 ng/mL for letrozole.

**Figure 2 ijms-25-10453-f002:**
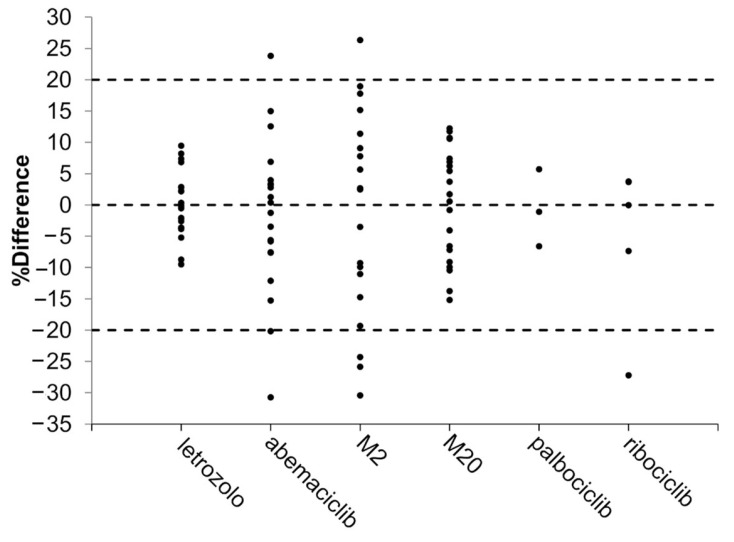
Percent difference (%Difference) of ECpla relative to actual Cpla obtained for each analyte. The accepted range (±20%) of %Difference is represented by dashed lines.

**Figure 3 ijms-25-10453-f003:**
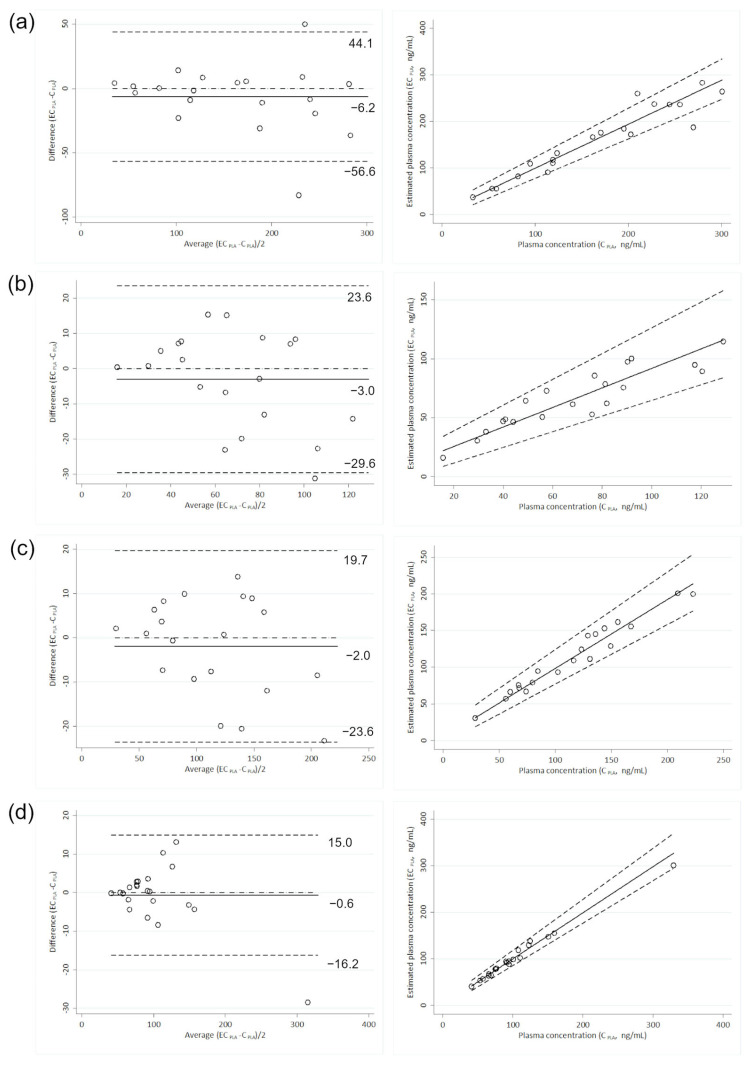
Relationship between actual and estimated plasma values of abemaciclib (**a**), M2 (**b**), M20 (**c**), and letrozole (**d**) based on Bland–Altman plot (left) and Passing–Bablok regression (right). The lower and upper limits of agreement (±1.96 SD of the bias) are reported as dashed lines in Bland–Altman plots and expressed as ng/mL, while the bias is reported as a solid line. In the Passing–Bablok regression, the 95% confidence interval is reported as dashed lines.

**Figure 4 ijms-25-10453-f004:**
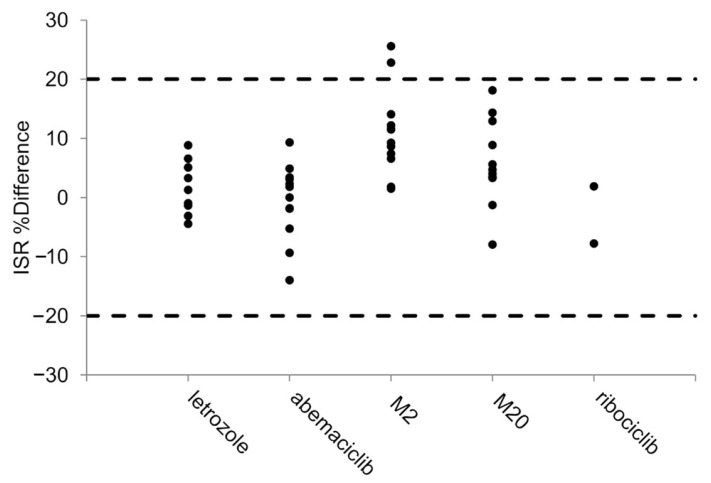
Incurred sample reanalysis. Dashed lines represent the accepted range (±20%) of percent difference (ISR %Difference).

**Figure 5 ijms-25-10453-f005:**
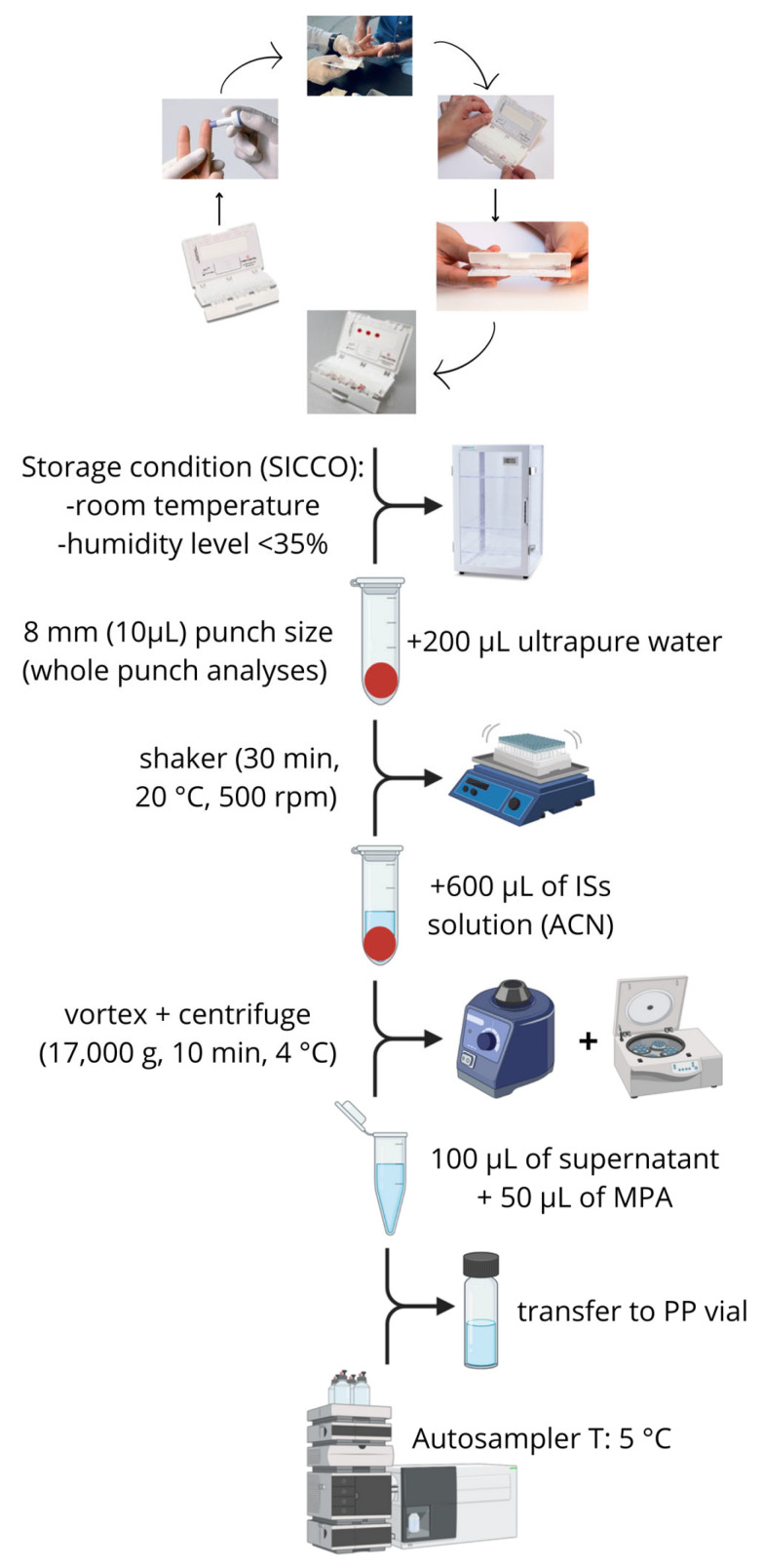
DBS sample process from blood collection by finger prick with the HemaXis device to extraction of the analytes.

**Figure 6 ijms-25-10453-f006:**
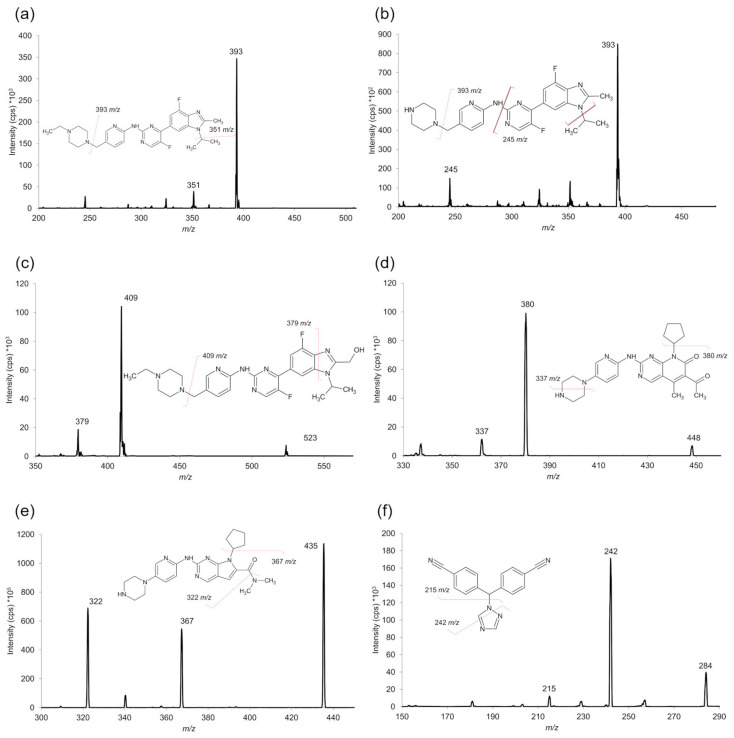
MS/MS mass spectra of abemaciclib (**a**), M2 (**b**), M20 (**c**), palbociclib (**d**), ribociclib (**e**), and letrozole (**f**) with chemical structures and identification of the main fragment ions.

**Table 1 ijms-25-10453-t001:** Between-run precision and accuracy data for letrozole, abemaciclib, M2, M20, palbociclib, and ribociclib.

Analyte	Nom Conc. (ng/mL)	Mean ± SD (ng/mL)	Acc%	CV%
letrozole	6.0	6.1 ± 0.4	102	6.0
	16.1	15.4 ± 0.6	96	3.8
	92.0	90.3 ± 2.0	98	2.2
	230.0	226.9 ± 5.4	99	2.4
abemaciclib	40.0	40.1 ± 2.9	100	7.3
	93.0	93.6 ± 5.7	101	6.1
	248.0	240.8 ± 6.3	97	2.6
	620.0	624.6 ± 23.7	101	3.8
M2	20.0	20.2 ± 2.1	101	10.5
	46.5	44.5 ± 2.4	96	5.3
	124.0	117.6 ± 5.7	95	4.8
	310.0	304.5 ± 15.6	98	5.1
M20	20.0	21.1 ± 2.1	106	10.2
	46.5	45.9 ± 3.4	99	7.5
	124.0	119.1 ± 4.8	96	4.0
	310.0	306.0 ± 17.8	99	5.8
palbociclib	6.0	5.9 ± 0.6	98	10.6
	16.1	15.5 ± 0.8	97	5.0
	92.0	90.9 ± 3.0	99	3.3
	230.0	225.2 ± 12.2	98	5.4
ribociclib	120.0	118.0 ± 11.2	98	9.5
	315.0	301.5 ± 10.1	96	3.4
	1800.0	1782.7 ± 46.1	99	2.6
	4500.0	4388.7 ± 166.4	98	3.8

**Table 2 ijms-25-10453-t002:** Accuracy and precision data of LLOQ and QC samples prepared at a Hct of 22 and 55%.

		Hct 22%			Hct 55%		
Analyte	Nom Conc. (ng/mL)	Mean ± SD (ng/mL)	Acc%	CV%	Mean ± SD (ng/mL)	Acc%	CV%
letrozole	6.0	6.5 ± 0.3	108	4.8	6.1 ± 0.1	102	1.8
	16.1	15.7 ± 0.3	97	2.1	15.6 ± 0.2	97	1.4
	92.0	91.3 ± 2.2	99	2.4	90.9 ± 1.4	99	1.5
	230.0	230.7 ± 3.3	100	1.4	227.5 ± 3.9	99	1.7
abemaciclib	40.0	41.4 ± 2.8	104	6.8	41.4 ± 2.7	103	6.6
	93.0	94.6 ± 2.1	102	2.2	89.9 ± 4.9	97	5.4
	248.0	252.5 ± 10.0	102	4.0	253.2 ± 9.5	102	3.7
	620.0	627.0 ± 19.4	101	3.1	608.8 ± 42.5	98	7.0
M2	20.0	22.1 ± 2.3	110	10,4	19.8 ± 2.4	99	12.1
	46.5	43.9 ± 3.1	94	7.1	43.5 ± 3.0	93	6.8
	124.0	122.2 ± 3.5	99	2.9	124.3 ± 4.6	100	3.7
	310.0	299.0 ± 16.1	96	5.4	301.3 ± 11.1	97	3.7
M20	20.0	22.2 ± 1.0	111	4.6	19.6 ± 1.7	98	8.5
	46.5	48.3 ± 3.1	104	6.4	45.5 ± 3.1	98	6.8
	124.0	128.0 ± 4.4	103	3.4	127.0 ± 5.2	102	4.1
	310.0	313.7±14.2	101	4.5	307.7±18.1	99	5.9
palbociclib	6.0	6.1 ± 0.8	102	12.5	5.6 ± 0.5	93	9.5
	16.1	15.9 ± 0.9	99	5.4	15.8 ± 0.8	98	4.8
	92.0	93.7 ± 2.0	102	2.1	91.7 ± 2.4	100	2.7
	230.0	237.0 ± 7.3	103	3.1	230.2 ± 6.9	100	3.0
ribociclib	120.0	128.5 ± 4.5	107	3.5	124.7 ± 6.8	104	5.5
	315.0	318.7 ± 10.6	101	3.3	304.2 ± 16.3	97	5.4
	1800.0	1835.0 ± 28.8	102	1.6	1756.7 ± 59.6	98	3.4
	4500.0	4420.0 ± 86.5	98	2.0	4357.7 ± 128.5	97	2.9

**Table 3 ijms-25-10453-t003:** Correction factors (CFs) applied for the conversion of DBS to plasma concentration for each analyte and relative percentage of samples that meet the equivalence criteria (% equivalence).

Analyte	N of Samples	CF (CV%)	CF from Non-Volumetric DBS [24]	% Equivalence
letrozole	22	0.9 (5%)	0.9	100%
abemaciclib	20	1.0 (14%)	-	90%
M2	20	1.4 (18%)	-	75%
M20	20	0.8 (9%)	-	100%
palbociclib	3	1.3 (7%)	1.4	100%
ribociclib	5	1.2 (5%)	1.4	80%

% equivalence: percentage of ECpla with a %diff with Cpla within ±20%.

**Table 4 ijms-25-10453-t004:** Comparison between actual and estimated plasma concentrations by means of Passing–Bablok regression, Bland–Altman, and Lin’s concordance correlation coefficient (Lin’s CCC) analyses.

Analyte	Passing–Bablok Regression				Lin’s CCC	Bland–Altman Analysis			
	Slope	95% CI	Intercept	95% CI		Bias	95% CI	ρ	*p*-Value
**letrozole**	1.0	0.9 to 1.1	0.9	−5.7 to 8.5	0.9901	−0.6	−4.2 to 2.9	−0.045	0.844
**abemaciclib**	0.9	0.8 to 1.1	5.0	−7.1 to 17.8	0.9444	−6.2	−18.3 to 5.8	−0.208	0.380
**M2**	0.8	0.7 to 1.1	9.0	−1.7 to 17.0	0.8850	−3.0	−9.4 to 3.3	−0.322	0.166
**M20**	0.9	0.8 to 1.1	4.2	−4.4 to 18.1	0.9743	−2.0	−7.1 to 3.2	−0.272	0.246

## Data Availability

All the relevant data are reported within the paper. For additional details, data are available on reasonable request to the corresponding author.

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
