# Peer review of "Quantification of Letrozole, Palbociclib, Ribociclib, Abemaciclib, and Metabolites in Volumetric Dried Blood Spots: Development and Validation of an LC-MS/MS Method for Therapeutic Drug Monitoring"

_ijms, 2024, doi:10.3390/ijms251910453_

Round 1

Reviewer 1 Report

Comments and Suggestions for Authors

Quantification of letrozole, palbociclib, ribociclib, abemaciclib, and metabolites in volumetric dried blood spots: development and validation of an LC-MS/MS method for therapeutic drug monitoring

E. Cecchin, M. Orleni, S. Gagno, M. Montico, E. Peruzzi, R. Roncato, L. Gerratana,

 S. Corsetti, F. Puglisi, G. Toffoli, E. Cecchin, and B. Posocco

The paper titled “Quantification of letrozole, palbociclib, ribociclib, abemaciclib, and metabolites in volumetric dried blood spots: development and validation of an LC-MS/MS method for therapeutic drug monitoring” by Cecchin et al. presents a targeted quantification protocol for monitoring drugs from dried blood samples.

The paper is clear in their goal. However, I felt the manuscript requires of an additional table which presents the following columns for the different metabolites: (1) Compound, (2) molecular weight, (3) Averaged RT, (4) Chemical formula, (5) CAS number, (6) Most important peaks in MS2 (total spectrum similarity % with databases), (7) simple/weight dot product, (8) reverse dot product, (9) isotopic-standard validated.

The goal of this table is to fulfill the “Proposed minimum reporting standards for chemical analysis” introduced by the Chemical Analysis Working Group (CAWG) Metabolomics Standards Initiative (MSI) [1].

[1] Summer, L. W., Amberg, A., Barrett, D., Beale, M. H., Beger, R., Daykin, C. A., ... & Hardy, N. (2007). Proposed minimum reporting standards for chemical analysis. Metabolomics, 3(3), 211-221.

Reviewer 2 Report

Comments and Suggestions for Authors

Lines of 72-73: Additional detailed explanation of what hematocrit effect means in DBS is required.

Lines of 73-76: Additional detailed explanation of “Volumetric DBS” is required. In particular, detailed explanation of how it differs from conventional DBS is required.

Additional explanation of the reasons for selecting target substances is required. Why were metabolites selected as targets only for abemaciclib? Pharmacological activity and related information for abemaciclib metabolites is required.

Line 147: Need explanation for “Hematocrit Effect”. What is the purpose of this test?

Line 176: Need specific information about clinical samples. Be transparent about patient demographics (disease status) and clinical trial approval information in the main text.

Line 204: Need additional information about the method and process for determining “Cpla”.

Figure 4: It is recommended that the X-axis and Y-axis of the graph be presented with full names instead of abbreviations.

Figure 4: The 95% confidence interval can also be displayed in the Passing-Bablock regression.

Lines of 278-299: It is recommended that the protocols be presented as visual “Figures” using schematic diagrams. This will help readers understand.

The narrative format presented in the current manuscript does not make it easy to grasp how actual dried blood samples were preprocessed to enable quantitative analysis of components. More detailed descriptions of the experimental procedures need to be added.

Lines of 305-307: Provide transparent information on the mobile phase gradient protocol used for simultaneous analysis.

Lines of 313-318: Provide the mass transitions of the analytes along with their structures. That is, provide information on where the product is generated within the parent structure.

Line of 309: Provide information on the ionization mode (positive or negative) of each analyte.

Figure S1: Does this mean that this is a comparison of methods for the extraction process of analytes in DBS? There is a lack of explanation in the manuscript for the interpretation of the results in Figure S1 (line 103).

Comments on the Quality of English Language

English should be rechecked by native speaker. 

Reviewer 3 Report

Comments and Suggestions for Authors

This study is aimed to develop a LC-MS/MS method for quantifying letrozole, abemaciclib and its metabolites, and further two CDK4/6 inhibitors in DBS. In comparison to some previous studies in this area, the HemaXis® device was used as alternative to VAMS for sample collection. Thus, the DBS-specific validation study and all the details that differ from the previous methods were described.

In addition, efforts were made in the optimization of sample treatment to reduce the hematocrite-based recovery bias.

The paper is interesting and well written. The main goals i.e. optimization of procedures for improving TDM in the breast cancer patients are important for both, clinical and scientific aspects.

To me the main adventure is elaboration of the methods which can be interchangeably used with determination in plasma, and for patients with different hematocrite levels.

Only minor editorial aspects could be improved, e.g.”

„..the final dilution step with FMA”; what is FMA?

„..with MeOH to obtain the WSs for both the calibrators”; what are WSs, working solutions?

Comments on the Quality of English Language

 Minor editing of English language required.

Round 2

Reviewer 2 Report

Comments and Suggestions for Authors

I have no further comments.

The revised manuscript is relatively well improved (according to reviewer's comments) than before.

Comments on the Quality of English Language

I have not felt significant English error.

Author Response

No further revision seems to be required by Reviewer 2.

Please let us know if other corrections are necessary.